# Continuous Time Random Walk with Correlated Waiting Times. The Crucial Role of Inter-Trade Times in Volatility Clustering

**DOI:** 10.3390/e23121576

**Published:** 2021-11-26

**Authors:** Jarosław Klamut, Tomasz Gubiec

**Affiliations:** Institute of Experimental Physics, Faculty of Physics, University of Warsaw, Pasteura 5, 02-093 Warsaw, Poland; Tomasz.Gubiec@fuw.edu.pl

**Keywords:** continuous time random walk, intertrade times, volatility clustering

## Abstract

In many physical, social, and economic phenomena, we observe changes in a studied quantity only in discrete, irregularly distributed points in time. The stochastic process usually applied to describe this kind of variable is the continuous-time random walk (CTRW). Despite the popularity of these types of stochastic processes and strong empirical motivation, models with a long-term memory within the sequence of time intervals between observations are rare in the physics literature. Here, we fill this gap by introducing a new family of CTRWs. The memory is introduced to the model by assuming that many consecutive time intervals can be the same. Surprisingly, in this process we can observe a slowly decaying nonlinear autocorrelation function without a fat-tailed distribution of time intervals. Our model, applied to high-frequency stock market data, can successfully describe the slope of decay of the nonlinear autocorrelation function of stock market returns. We achieve this result without imposing any dependence between consecutive price changes. This proves the crucial role of inter-event times in the volatility clustering phenomenon observed in all stock markets.

## 1. Introduction

In many physical, biological, and economic systems we can identify elementary events occurring irregularly in time. Additionally, the times between those events can be interdependent in a non-trivial manner, which can lead to complex behavior. Therefore, it is no surprise that point processes are of high interest to researchers and their applications are widely studied [1,2]. Two of the most popular models are autoregressive conditional duration (ACD) [3] and the Hawkes model [4,5]. The canonical versions of both models include short-range dependencies (for ACD see [3,6,7,8,9,10,11]; for Hawkes see [12,13,14,15,16,17,18,19,20]). Both of them, however, have been extended to describe long-range memory (for ACD see [21,22,23,24,25,26,27,28,29,30,31]; for Hawkes see [32,33,34,35,36,37,38,39,40,41,42]).

Real-world stochastic processes have numerous features which can be associated with elementary events. For instance, in the transaction data from a stock market we observe the events—the transactions occurring in specific moments—and their features: the price and volume of each transaction. Inter-trade times from stock market transaction data are a perfect example of a point process. However, in order to describe the price of transactions, which we do below, one must go beyond the framework of point processes, which does not incorporate features of the elementary events. A natural generalization is the continuous-time random walk (CTRW).

The CTRW was the first proposed formalism to describe the dynamics of a variable changing its value in unevenly spaced points in time. Point processes extended to fit this phenomenon are called marked point processes [16]. Moreover, the distribution of time intervals between those points can be arbitrary. This formalism was introduced in 1965 by Montroll and Weiss [43] and since then it has been applied in a broad range of fields, ranging from astrophysics to economics and the social sciences. For a detailed review, see [44]. In the canonical CTRW, both increments of the observed process and waiting times (inter-event times) are i.i.d. random variables. An exemplary trajectory of such a process is shown in Figure 1.

All kinds of random walks, starting with normal diffusion, through anomalous diffusion (both subdiffusion and superdiffusion) to Levy flights, can be described within the CTRW formalism. This can be achieved by using specific distributions of waiting times or increments (especially with heavy tails) and by considering memory in waiting times, increments, or coupling between them. The CTRW models with correlated increments were initially proposed to study lattice gases [45,46,47]. More recently, they have been used to model high-frequency financial data [48,49,50,51,52,53,54,55,56,57,58,59,60]. On the other hand, CTRW models with correlated waiting times are not well-studied. With the exception of a few recent attempts [52,61,62], these models have not been analyzed nor used to model empirical data. This fact is surprising in light of the recent popularity of point processes such as ACD and the Hawkes process. The aim of this work is to fill this gap. We propose a new CTRW model which incorporates dependencies of inter-event times. Our intention is to model long-range memories in the sequence of waiting times, an aim inspired by numerous empirical examples [63,64,65,66,67,68,69]. Our model is simple yet general enough to explain the properties of empirical data. That makes it a perfect candidate for future applications and a relevant reference point for future work.

The paper is organized as follows. In Section 2, we present the motivation behind the model, with correlated waiting times based on financial data. Next, in Section 3 we propose a way to include dependencies between the waiting times, in particular the long-range memory. In Section 4, we solve the CTRW model with correlated waiting times by calculating its propagator, moments, and the autocorrelation function (ACF) of increments. We then fit our model to tick-by-tick transaction data from the Warsaw Stock Exchange in Section 5. Finally, we provide a summary of our work in Section 6. Two appendices at the end provide a clarification of the mathematical methods that we have used.

## 2. Motivation

Models with interdependent waiting times are used to describe electron transfer [63], the firing of a single neuron [64], interhuman communication [65], and the modeling of earthquakes [66,67,68,69]. An excellent example of a process with correlated inter-event times that we will describe in this manuscript is tick-by-tick transaction price data from the stock market [70]. These data are very convenient to use, as they are of high quality and easily accessible in large amounts.

Firstly, let us recall two basic stylized facts observed in the majority of stock markets [71].

In the ACF of time-dependent log-returns, we observe short-term negative autocorrelation.However, we observe slowly decaying positive autocorrelation for the ACF of absolute values of time-dependent log-returns.

The latter is considered to be reminiscent of the volatility clustering phenomenon.

Of course, these are not the only or the most significant stylized facts, but these two do not directly depend on the log-return distribution. The list should also contain the broad distribution of log-returns [72]; multi-fractality [73,74]; universal scaling of the distribution of times between large jumps [75,76]; and the slow, power-law decay of the correlation between these times. We will further discuss the latter in this manuscript. Usually, the CTRW models used to describe high-frequency stock market data consider waiting times $\Delta t_{n}$ as inter-transaction times, and process increments $\Delta x_{n}$ as logarithmic returns between consecutive transactions. Taking into account the so-called bid-ask bounce phenomenon allows CTRW processes to reproduce the first stylized fact of short-term negative autocorrelation [58,77,78]. In this type of models, waiting times $\Delta t_{n}$ are i.i.d. variables and only the dependence between $\Delta x_{n}$ and $\Delta x_{n-1}$ is considered. Unfortunately, models considering only this type of dependencies turned out to be unable to describe the time ACF of absolute values of price changes [60]. Technically, it is possible to obtain a CTRW model reproducing both stylized facts, but it requires a power-law waiting-time distribution ψ(Δt). However, this solution is not satisfying as we can obtain waiting-time distribution directly from the empirical data of inter-transaction times. It turns out that this distribution is far from a power-law one [58]. These results suggest that the source of the second stylized fact is not in the distributions of increments h(Δx) and waiting times ψ(Δt), but in the dependence between consecutive Δx and Δt values.

Let us start with an empirical analysis of the step ACF of series $\Delta t_{n}$ and $|\Delta x_{n}|$. We observe approximately power-law memories in waiting times and absolute values of price changes; see Figure 2a. For a lag (in the number of steps) ≲3, the autocorrelation of $|\Delta x_{n}|$ is higher than the autocorrelation of $\Delta t_{n}$, but for a lag >3 it is otherwise. This result suggests that in the limit of long times, the dependence between waiting times may be more critical than dependence between price changes. To verify this hypothesis we perform a shuffling test. We compare the time ACF of price changes’ absolute values for four samples of time series. The first one is the original time series of tick-by-tick transaction data. The second time series keeps the price changes $\Delta x_{n}$ in the original order but shuffles the order of waiting times $\Delta t_{n}$. This way, we obtained a time series keeping all dependencies between price changes $\Delta x_{n}$, but without any dependencies between waiting times $\Delta t_{n}$. In the third time series, we kept the original waiting times $\Delta t_{n}$ but shuffled the price changes $\Delta x_{n}$. In the last, fourth time series, both $\Delta t_{n}$ and $\Delta x_{n}$ were shuffled. Let us emphasise that all four time series have the same, unchanged distributions $\psi (\Delta t_{n})$ and $h(\Delta x_{n})$. The results are shown in Figure 2b. As expected, we observe the slow, almost power-law decay of the time ACF for the first empirical time series. Surprisingly, removing dependencies between waiting times does not change the time ACF in the limit of t→0, but significantly increases its slope of decay in the long-term. On the other hand, removing dependencies between price changes decreases the time ACF, dividing it by an almost constant factor but does not change the slope of the decay. The removal of all dependencies still leads to a positive time ACF, resulting from the non-exponential empirical distribution of waiting times.

The empirical observations presented above convinced us that it is necessary to consider long-range dependencies between waiting times within CTRW to reproduce the slowly decaying ACF of price changes’ absolute values observed in the financial data.

Please note that in Figure 2, we analyzed the step ACF for lags up to 100 and the time ACF for times up to 1000 s. The procedure used to estimate the time ACF was presented in [58] and is a modification of the classical slotting technique introduced in [79]. Such limits were chosen due to the length of trading sessions (around 8 h or 1000 trades). Unfortunately, these limits are not long enough to detect power-law dependencies. The only way to increase these limits is by joining all sessions into one sequence. In this procedure, we merge the end of one session with the beginning of the following one (we omit overnight price changes). These two periods of the sessions are different, as we observe intraday activity in financial data [80]. The session begins with short inter-transaction times and a high standard deviation of price changes. Usually, up to the middle of the session, average inter-trade times increase, and the standard deviation of price changes decreases. The situation reverts again close to the end of the session. This phenomenon is called the *lunch effect* [81]. We use the canonical method to remove intraday non-stationarity by dividing each waiting time by the corresponding average waiting time, depending on the time that has elapsed since the beginning of the session for each day of the week separately [82,83]. The comparison of the step ACFs of waiting times for non-stationarized and stationarized data is presented in Figure 3a. As a result of this procedure, we obtain the power-law decay over four orders of magnitude of lag. In Figure 3b, we present the time ACF of price changes’ absolute values for stationarized data, which also exhibit power-law decay over four orders of magnitude of time lag. It is now reasonable to ask what the relationship is between the decay exponents of these autocorrelations. Fortunately, the model studied in this paper gives a strict answer to this question.

## 3. Process of Waiting Times

Let us now focus on the sequence of inter-transaction times $\Delta t_{1},\Delta t_{2},\ldots ,\Delta t_{n},\ldots$. We are now looking for the point process to describe this series, which will be suitable for use in CTRW. For this reason, we need analytically solvable models. Moreover, we would like to use the empirical distribution of inter-event times $\psi (\Delta t_{n})$ and observe the power-law step ACF, as shown in Figure 3a. Even these two simple conditions exclude ACD models and Hawkes processes from our considerations. We are not interested in ACD models, as the power-law ACF can be obtained only within the fractional extension. In the Hawkes process, both the waiting time distribution and autocorrelation depend on the memory kernel [15,84]. Therefore, they cannot be set independently. As the Hawkes process is defined solely by its kernel, both waiting time distribution and autocorrelation depend on it. Thus, it would be difficult (if it is possible at all) to reproduce both empirical WTD and ACF at the same time. This feature of the Hawkes process hampers its use in the description of empirical data.

As the solution to our search, we propose a simple point process in which waiting times Δt are repeated. In a very general sense, our proposition can be interpreted as a discretized version of CTRW, adapted to the role of the point process. Let us briefly describe this analogy. Within the canonical CTRW, values of the process are represented by a spatial variable, and the time is continuous. The spatial variable remains constant for a given period of continuous waiting time. Now, we define the point process by the series of waiting times. Here, the number of repetitions $\nu_{i}$ of the same value of waiting time is the analog of waiting time in the canonical CTRW. The exemplary realization of such an adapted process of waiting times is shown in Figure 4.

We require the waiting times $\Delta t_{n}$ (values of the process in the discrete subordinated time *n*) to come from the distribution $\psi (\Delta t_{n})$ ($\Delta t_{n}>0$), with a finite mean 〈Δt〉. We define $\nu_{i}$ as the number of repetitions of the same waiting times (drawn independently for each series of repetitions). Let $\nu_{i}$ be the i.i.d. random variables with the distribution $\omega (\nu_{i})$. In general, it can be any distribution, but to recreate the power-law step ACF of waiting times we will focus on a fat-tailed distribution with a finite first moment 〈ν〉. In particular, we use the zeta distribution with parameter ρ
(1)$$\omega_{\rho }\left(k\right)=k^{-\rho }/\zeta \left(\rho \right);\;\zeta \left(\rho \right)=\sum_{i=1}^{\infty }i^{-\rho },\;\rho >1,$$
where ζ(ρ) is Riemann’s zeta function. Its expected value is equal to $\langle \omega \rangle =\frac{\zeta (\rho -1)}{\zeta (\rho )}$ for ρ>2 and the variance is finite for ρ>3. The cumulative distribution function is given by $\frac{H_{k,\rho }}{\zeta (\rho )}$, where $H_{k,\rho }=\sum_{i=1}^{k}i^{-\rho }$ is the generalized harmonic number. Let us introduce $\Omega \left(k\right)=\sum_{i=k}^{\infty }\omega \left(i\right)$ as a sojourn probability. We have $\Omega \left(k\right)=1-\frac{H_{k-1,\rho }}{\zeta (\rho )}$ for the zeta distribution.

We define a soft propagator of the process of times $P(\Delta t;n|\Delta t_{0},0)$, which is the conditional probability density that the waiting time, which was initially (at n=0) in the origin value ($\Delta t=\Delta t_{0}$), is equal to Δt after *n* steps. The soft propagator can be expressed by
(2)$$P\left(\Delta t;n|\Delta t_{0},0\right)=\delta \left(\Delta t-\Delta t_{0}\right)\Omega^{\mathrm{first}}\left(n\right)+\left[1-\Omega^{\mathrm{first}}\left(n\right)\right]\psi \left(\Delta t\right),$$
where $\Omega^{\mathrm{first}}\left(n\right)$ is the sojourn probability obtained from $\omega^{\mathrm{first}}\left(n\right)$, which is the stationarized distribution of the repetition of the first waiting time:(3)$$\begin{array}{cc} \omega^{\mathrm{first}}\left(n\right) & =\frac{\sum_{n^{\prime }=1}\omega \left(n+n^{\prime }\right)}{\sum_{n^{\prime \prime }=0}\sum_{n^{\prime }=1}\omega \left(n^{\prime \prime }+n^{\prime }\right)}=\frac{\sum_{n^{\prime }=1}\omega \left(n+n^{\prime }\right)}{\sum_{n=1}n\omega \left(n\right)}=\frac{\sum_{n^{\prime }=n+1}\omega \left(n^{\prime }\right)}{\langle \omega \rangle },\\ \Omega^{\mathrm{first}}\left(n\right) & =\frac{\sum_{i=n}\sum_{n^{\prime }=i+1}\omega \left(n^{\prime }\right)}{\langle \omega \rangle }=\frac{\sum_{i=1}i\omega \left(i+n\right)}{\langle \omega \rangle }=\frac{\langle \omega \rangle -n\Omega \left(n+1\right)-\sum_{i=1}^{n}i\omega \left(i\right)}{\langle \omega \rangle }. \end{array}$$
The first term of the right-hand side of Equation (Equation 2) is the probability that the process value will stay constant (equal $\Delta t_{0}$) after *n* jumps. The second term indicates that there will be a process value jump with probability $1-\Omega^{\mathrm{first}}\left(n\right)$, so new process values will be completely independent, drawn from the distribution ψ(Δt).

Restricting ourselves to ω(n) in the form of the zeta distribution, we can obtain
(4)$$\Omega^{\mathrm{first}}\left(n\right)=1-\frac{n}{\langle \omega \rangle }+\frac{nH_{n,\rho }}{\zeta (\rho -1)}-\frac{H_{n,\rho -1}}{\zeta (\rho -1)},$$
and hence the propagator given by Equation (Equation 2). The step autocovariance of waiting times $\Delta t_{n}$ can be expressed as
(5)$$\mathrm{cov}\left(n\right)=\langle \Delta t_{i}\Delta t_{i+n}\rangle -\langle \Delta t_{i}\rangle \langle \Delta t_{i+n}\rangle =\langle \Delta t_{i}\Delta t_{i+n}\rangle -{\langle \Delta t\rangle }^{2},$$
where symbol 〈…〉 means taking the average. Note that $\Delta t_{i+n}=\Delta t_{i}$ with probability $p=\Omega^{\mathrm{first}}\left(n\right)$. With probability 1−p, the $\Delta t_{i}$ is independent. This leads to
(6)$$\mathrm{cov}\left(n\right)=p\langle \Delta t^{2}\rangle +\left(1-p\right){\langle \Delta t\rangle }^{2}-{\langle \Delta t\rangle }^{2}=\sigma_{\Delta t}^{2}p=\sigma_{\Delta t}^{2}\Omega^{\mathrm{first}}\left(n\right).$$
We are interested in the asymptotic form of autocorrelation for n≫1. We can use following approximation (Theorem 12.21 from [85])
(7)$$\zeta \left(\rho \right)-H_{n,\rho }\approx \frac{n^{1-\rho }}{\rho -1}.$$
Finally, we obtain the normalized step ACF
(8)$$\mathrm{corr}\left(n\right)=\frac{\mathrm{cov}(n)}{\mathrm{cov}(0)}\approx \frac{n^{-(\rho -2)}}{\zeta (\rho -1)(\rho -2)(\rho -1)}.$$
The step ACF of waiting times decays like a power-law and the decay exponent is ρ−2. It is worth emphasizing that even considering only ρ>2, required for the existence of a finite average number of repetitions, we can obtain any value of the decay exponent.

## 4. The Primary Process

Now we are ready to define the primary CTRW process with repeating waiting times. This process is characterized by two key properties:changes of the process value $\Delta x_{n}$ are i.i.d. random variables from the distribution h(Δx), with finite variance $\sigma_{x}^{2}$ (and thus finite first two moments $\mu_{1}$ and $\mu_{2}$),waiting times $\Delta t_{n}$ come from the process described in Section 3.

Note that we do not assume any dependence within the series of consecutive changes of the process value $\Delta x_{1},\Delta x_{2},\ldots ,\Delta x_{n}$. We do not make any further assumptions about the shape of distributions h(Δx). The memory in this process is present only in the sequence of waiting times.

Let us start the analysis of the properties of this process with the following observation. As the changes $\Delta x_{n}$ are independent, the changes above any given threshold occur independently. Knowing the result in (Equation 8), we can calculate the autocorrelation of the series of inter-occurrence times between changes above or below any threshold. The details of the derivation are presented in Appendix B. It turns out that we also obtain power-law decay with the exponent −(ρ−2), the same as in (Equation 8).

Moreover, we managed to obtain the soft propagator of the primary CTRW process and the characteristics derived from it. The details of calculations can be found in Appendix A. Here we present selected results, namely, the first two moments and the time autocorrelation of changes, in the limit of long times (t→∞). We consider analytical terms ($t,t^{2},t^{3},\ldots$) and the most significant power-law term when ρ is non-integer.

Using results from Appendix A, the first moment of the process for t→∞ can be approximated as
(9)$$m_{1}\left(t\right)=L^{-1}\left[-i\frac{\partial \tilde{P}\left(k;s\right)}{\partial k}{|}_{k=0}\right]\left(t\right)\approx \frac{\mu_{1}}{\langle \Delta t\rangle }t+\mu_{1}\frac{\alpha \{\psi \}}{\Gamma (4-\rho )}t^{3-\rho },\;\rho \in \left(2;4\right),$$
where $L^{-1}\left[\cdot \right]\left(t\right)$ is the inverse Laplace transform, $\tilde{P}\left(k;s\right)$ is the propagator of the process in the Fourier–Laplace domain, Γ(·) is Euler’s gamma function, and α{ψ} is a complex functional of ψ, which has to be calculated separately for each ψ. The most important term is typical, linear behavior, but we observe an additional power-law term. The second moment can be written in the form
(10)$$\begin{array}{cc} m_{2}\left(t\right) & =L^{-1}\left[-\frac{\partial^{2}\tilde{P}\left(k;s\right)}{\partial k^{2}}{|}_{k=0}\right]\left(t\right)\\ \; & \approx \mu_{1}^{2}{\left(\frac{t}{\langle \Delta t\rangle }\right)}^{2}+\sigma_{x}^{2}\frac{t}{\langle \Delta t\rangle }+\mu_{1}^{2}\beta \left\{\psi \right\}\frac{t}{\langle \Delta t\rangle }+\mu_{1}^{2}\frac{\gamma \{\psi \}}{\Gamma (5-\rho )}t^{4-\rho },\;\rho \in \left(2;5\right), \end{array}$$
where β{ψ},γ{ψ} are complex functionals of ψ, which have to be calculated separately for each ψ. From the first two moments of the process, we calculate the process variance (still considering only analytical and the most important power-law term)
(11)$$\sigma^{2}\left(t\right)=m_{2}\left(t\right)-m_{1}^{2}\left(t\right)\approx \left(\sigma_{x}^{2}+\mu_{1}^{2}\beta \left\{\psi \right\}\right)\frac{t}{\langle \Delta t\rangle }+\mu_{1}^{2}\frac{\gamma \{\psi \}}{\Gamma (5-\rho )}t^{4-\rho },\;\rho \in \left(2;5\right).$$
It is worth mentioning that for variance the power-law term from the second moment is more important than the power-law term from the first moment. We can observe normal diffusion for ρ>3. However, there is superdiffusion in the case of ρ∈(2;3). We obtain ballistic diffusion in the limit ρ→2.

Having the first two moments, one can calculate velocity ACF, which is equivalent to normalized ACF of changes for fixed sampling for the stationary process
(12)$$C\left(t\right)=\frac{1}{2}\frac{\partial^{2}m_{2}\left(t\right)}{\partial t^{2}}-{\left(\frac{\partial m_{1}\left(t\right)}{\partial t}\right)}^{2}\Rightarrow C\left(t\right)\approx \mu_{1}^{2}\frac{1}{\Gamma (3-\rho )}\kappa \left\{\psi \right\}t^{2-\rho },$$
where $\kappa \left\{\psi \right\}=\left(\frac{\gamma \{\psi \}}{2}-\frac{2\alpha \{\psi \}}{\langle \Delta t\rangle }\right)$, for ρ∈(2;4). In the limit of t→∞ and $\mu_{1}\neq 0$ we observe a power-law decay of ACF with the exponent ρ−2. In the case of $\mu_{1}=0$, it can be proven that this exponent is ρ−1, so the decay is faster (Equation 17).

It is crucial to emphasize that in Equations (Equation 9)–(Equation 12) for ρ exceeding the mentioned range, there is still a power-law term with the same dependence on $\mu_{1}$ and the same time exponent. However, the dependence of the amplitude on ρ takes a different, more complex form.

## 5. Empirical Results

We use the constructed process to investigate the role of correlated inter-trade times in the volatility clustering effect. We consider this process as a toy model, describing high-frequency financial data. The value of the process represents the logarithm of the stock price. We can treat transactions as events that change the price. Therefore, the inter-transaction times correspond to waiting times in our model. The jumps represent the difference in the logarithmic prices of consecutive transactions, which are logarithmic returns [52].

The CTRW formalism allows us to obtain the autocorrelation of price returns. Moreover, the same formalism can be used to obtain the nonlinear ACF of absolute increments. This can be achieved by using different jump distributions h(Δx). To model the process of price changes in time, we should use the symmetric distribution h(Δx), as the empirical distribution of returns is symmetrical. As a result, we obtain the vanishing mean $\mu_{1}=0$ and the quickly decaying ACF of returns. To derive the nonlinear ACF of absolute returns, we define the new CTRW process, and by calculating its linear ACF, we obtain the nonlinear ACF of price increments. Following [60], if as h(Δx) we use only the positive half of the previous distribution multiplied by 2, we deal with the case of non-zero drift and obtain an artificial, monotonically increasing process. As $\mu_{1}\neq 0$, we obtain the slow power-law decay of the autocorrelation of absolute returns, as in the empirical results presented as a solid black line in Figure 2b.

Since we assumed only one type of memory in our model, introduced by the distribution ω(ν), we cannot expect that the model will be able to reproduce exact values of the empirical nonlinear ACF of the absolute returns. The model, however, should be able to reproduce its slope (as in Figure 2b, in which the green dash-dotted line reproduces the slope of the solid black line). The theoretical slope is obtained analytically and is equal 2−ρ. It is worth emphasizing that the slope does not depend on the distribution of price changes h(Δx) or waiting times ψ(Δt) and is fully determined by the single parameter ρ, characterizing the distribution ω(ν). This fact significantly simplifies the comparison with the empirical data, as we are required to estimate only one parameter ρ. On the other hand, the assumption of repeated waiting time is a technical method introducing memory. We cannot expect to observe such a phenomenon in the empirical time series. The parameter ρ is a measure of the memory present in the sequence of consecutive waiting times. Therefore, we estimate this parameter using the slope of the step ACF of waiting times, which is equal to 2−ρ in the model. It is a surprising and potentially essential fact that the exponent of the decay of the nonlinear time ACF is the same as in the step ACF of waiting times. This result motivates us to compare these two values for empirical financial data. Of course, in the empirical data we also observe a long-term positive step ACF of |Δx|, which was not included in our model. Therefore, we can expect that the slope of time ACF of |Δx| should be slightly higher than the slope of the step ACF of Δt. Since a long-term nonlinear autocorrelation is usually interpreted as a reminiscence of the volatility clustering phenomenon, it is interesting to check what part of the observed volatility clustering effect can be explained only by memory between inter-trade times. We present the results for the five most traded stocks from the Warsaw Stock Exchange in Table 1 (ordered by the number of transactions), with the average inter-trade time not being greater than 30 s.

We see that our model can estimate the slope of time ACF with an accuracy of around 10%. Moreover, our model can successfully reproduce the power-law decay of the autocorrelation of inter-occurrence times between changes below or above any given threshold reported in [75,76]. Please note that the decay exponent predicted by our model −(ρ−2), with empirical values presented in the Table 1, is close to 0.31, as reported in [76].

## 6. Conclusions

We introduced a new continuous-time random walk (CTRW) model with long-term memory within a sequence of waiting times. We use a simple model of repeating waiting times instead of commonly-used point processes such as the ACD and the Hawkes process. Despite its simplicity, our model of repeating waiting times has a few valuable properties. It is stationary, can be treated analytically, and the distribution of waiting times and memory in its series can be set independently.

As we observe many phenomena with dependencies between waiting times, possible applications of this family of CTRW models go beyond the exemplary application presented here.

However, in this manuscript, we applied the proposed model to describe high-frequency financial time series. We asked ourselves which commonly known properties of the financial time series can be reproduced by the long-term memory introduced in our model, only by means of the repeating waiting times. We have to emphasize that part of these properties, known as stylized facts, depend on the waiting time distribution ψ(Δt) and price change distribution h(Δx). As we are not trying to study the general ability of continuous-time random walk to describe the high-frequency financial time series, we have not studied the broad distribution of log-returns [72], multi-fractality [73,74], or universal scaling of the distribution of times between large jumps [75,76]. We have analyzed the decay of the nonlinear time autocorrelation function of log-returns and the decay of the step autocorrelation function of times between large jumps. Although we considered only memory in a sequence of waiting times, we managed to show that long-term dependencies in waiting times are crucial in explaining the volatility clustering effect and results in the power-law decay of both measures mentioned above.

Our results indicate that the dependence between consecutive price changes is not the primary carrier of long-range memory in the volatility clustering phenomenon. To verify these results, we conducted another simulation. We prepared autocorrelated series of waiting times according to the Fourier filtering method (for example, described in [86]). Similarly, as in our model, both the slopes of the step ACF of WTs and the time ACF of absolute returns were the same. This verification confirms our conclusion and indicates that it is general, independently of the origin of the autocorrelation between inter-trade intervals.

## Figures and Tables

**Figure 1 entropy-23-01576-f001:**
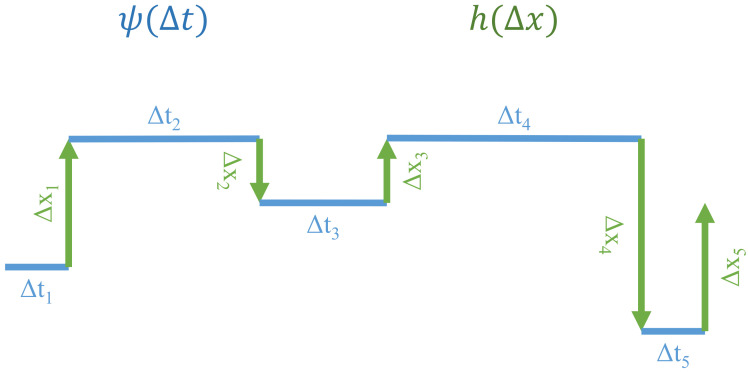
The example trajectory of the continuous-time random walk (CTRW), consisting of jumps of process values $\Delta x_{n}$ preceded by waiting times $\Delta t_{n}$. In the canonical CTRW, $\Delta t_{n}$ and $\Delta x_{n}$ are i.i.d. random variables drawn from the distributions $\psi (\Delta t_{n})$ and $h(\Delta x_{n})$, respectively. In this paper, we consider the CTRW model with long-term dependence in the series of waiting times $\Delta t_{1},\Delta t_{2},\ldots ,\Delta t_{n}$.

**Figure 2 entropy-23-01576-f002:**
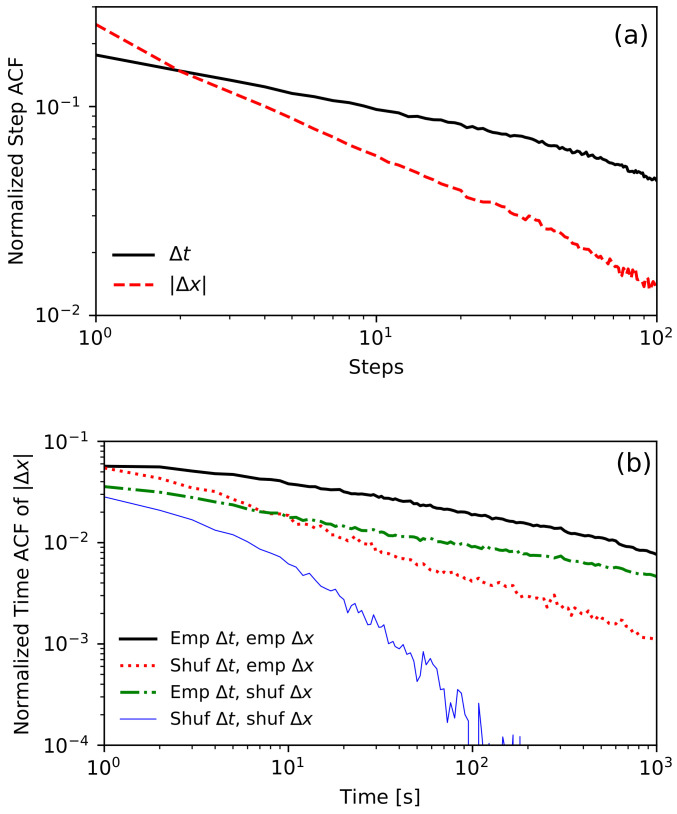
Figure 2 and Figure 3 were prepared using transaction data for KGHM (one of the most liquid Polish stocks) from period of January 2013 to July 2017. Both figures are on a log-log scale. (**a**) The plot of normalized empirical step ACF of Δt and |Δx|. Both functions decay like a power-law. For lag = 1, the autocorrelation of |Δx| is higher. However, it decays faster, and for long times the memory in waiting times is stronger. (**b**) The plot of the normalized time ACF of |Δx| for four time series. The presented lines are for empirical data (thick black), empirical price changes, and intra-daily shuffled waiting times (dotted red); intra-daily shuffled price changes and empirical waiting times (dash-dotted green); and intra-daily independently shuffled price changes and waiting times (thin blue). Considering only empirical dependencies of waiting times reproduces the ACF, which decays with almost the same slope as the empirical one.

**Figure 3 entropy-23-01576-f003:**
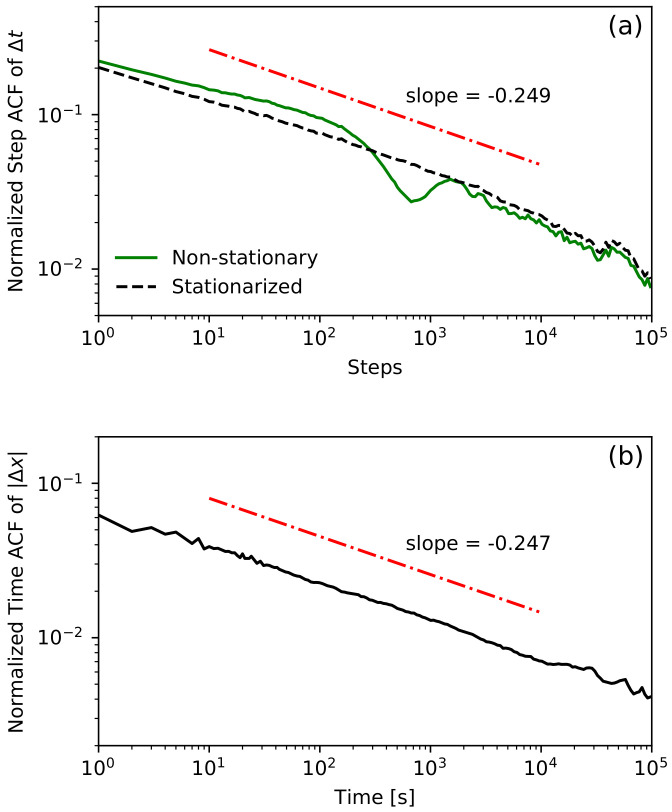
All intraday data (waiting times and corresponding price changes) are joined into one data set. (**a**) The plot shows the normalized step ACF of Δt for non-stationary and stationarized cases. The stationarizing procedure is described in the main text. (**b**) The plot of the normalized time ACF of |Δx| with stationarized waiting times. Both stationarized autocorrealations decay like a power-law with similar slope.

**Figure 4 entropy-23-01576-f004:**
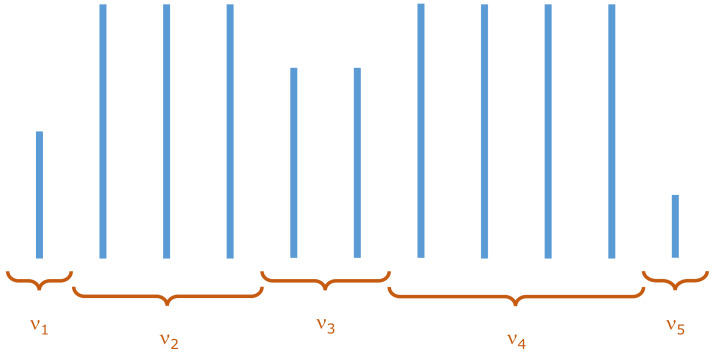
The example realization of the process of waiting times, the values of which correspond to the waiting times $\Delta t_{n}$ of the point process used in the primary CTRW process. Process values $\Delta t_{1},\Delta t_{2},\ldots ,\Delta t_{n}$ come from the values $\Delta t^{1},\Delta t^{2},\ldots ,\Delta t^{k}$ repeated $\nu_{1},\nu_{2},\ldots ,\nu_{k}$ times, respectively. Number of repetitions $\nu_{i}$ are drawn from the distribution $\omega (\nu_{i})$. In the example above: $\nu_{1}=1,\nu_{2}=3,\nu_{3}=2,\ldots$ and $\Delta t_{1}=\Delta t^{1},\Delta t_{2}=\Delta t_{3}=\Delta t_{4}=\Delta t^{2},\Delta t_{5}=\Delta t_{6}=\Delta t^{3},\ldots$.

**Table 1 entropy-23-01576-t001:** Table with fitted slopes of the empirical stationarized step ACF of waiting times and the time ACF of price changes’ absolute values for the five most liquid stocks from the WSE. The time ACF slopes are close to the corresponding step ACF slopes. The analysis was performed on the tick-by-tick market data from the public domain database [70]. The data covers the period from 3 January 2013 to 14 July 2017. For instance, the data set for KGHM contains 3,096,625 transactions.

Company	Step ACF Δt Slope	Time ACF |Δx| Slope
KGHM	−0.25±0.04	−0.25±0.02
PKOBP	−0.33±0.08	−0.30±0.02
PZU	−0.26±0.03	−0.28±0.04
PGE	−0.33±0.07	−0.36±0.03
PEKAO	−0.33±0.04	−0.37±0.04

## Data Availability

No new data were created or analyzed in this study. Data sharing is not applicable to this article.

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
