# Peer review of "Continuous Time Random Walk with Correlated Waiting Times. The Crucial Role of Inter-Trade Times in Volatility Clustering"

_entropy, 2021, doi:10.3390/e23121576_

Round 1

Reviewer 1 Report

The 1965 CTRW paper lay dormant until the Scher-Lax and Scher-Montroll papers of the 70's.  These papers showed the power of the CTRW process for long tailed distributions.  Afterwards an explosion of works followed and recounted in the 50 year anniversary issue, the authors reference 44.  The authors show there is still much richness to be mined in the CTRW context.  They introduce correlated blocks of waiting times in a new aspect to the model.  They obtain analytic results for moments and correlations and find applications to high speed financial trading.  The results are new and a welcome addition to the CTRW family.  No doubt their model will attract even further generalizations.

Author Response

We would like to thank the Reviewer for these comments.

Reviewer 2 Report

The paper presents a model for addressing the interevent, or sojourn, time intervals in the Continuous Time Random Walk framework. The model introduces memory by allowing many successive sojourns to have the same amplitude which, in turn, provides a sort of long-range memory. The model is applied to tic-by-tic financial data obtained from a sample of companies belonging to the Warsaw Stock Exchange. The authors, in a somewhat indirect way, suggest that the model may be suitable to explain the volatility cluster effect as well as to describe a slow decay in the autocorrelation function.

The manuscript is interesting and mathematically sound. It presents new research and, although a little intricate in its exposition, it is well written. In my opinion the manuscript deserves publication in Entropy.

Author Response

(The authors gave the same response as above.)

Reviewer 3 Report

Inspired by certain traits observed in financial time series, the authors present in this manuscript a model of Continuous-Time Random Walk (CTRW) for the evolution of a process $x(t)$ where consecutive instances of the inter-event times can take exactly the same value. With this premise, the authors center their efforts in the analysis of the autocorrelation function (ACF) of the waiting times and the increments of the process. This is one of the major weakness of the manuscript, the absence of a more comprehensive analysis of the whole model, beyond the first two moments or the ACFs.

The second weakness is that, as the authors confess at the end of section 5, the model is a technical artifact introduced for mathematical convenience. This diminishes the relevance of the model. 

The third one is a certain lack of consistency in the notation, with implications in the right understanding of the complete CTRW process. In most of the paper, e.g., in Fig. 4 or in Appendix A, $\Delta t_n" denotes the n-th outcome of the random variable whose probability density function (PDF) is represented by $\psi$. Accordingly, the number of consecutive replicas of random variate $\Delta t_n$ is symbolized by $\nu_n$. Therefore, within this interpretation, unless the PDF $\psi$ has atoms, the probability that $\Delta t_{i+n}=\Delta t_{i}$ is zero, not $p$ as stated after Eq. (5). Indeed, in this equation and related expressions, $\Delta t_n$ represents each individual waiting time, counting the repetitions as different sojourns. Summing up, in this second interpretation, $n$ counts the number of sojourns, whether they are distinct or not. Associated to that, and according to Eq. (A4), $\Delta x_n$ represents the change that suffers the CTRW after every individual sojourn. Definitively, the authors should use a better notation.

Besides that, I have some comments with a mixed level of significance:

-The term "trajectory" when describing the mechanism that generates the waiting times is very, very misleading. The model defines a point process on the basis of the drawn of random intervals with a random number of replicas.

-Indeed, the authors tends to identify "the process", not with the CTRW itself but with the point process alone. 

-The expression "unknown functional of $\psi$" that is used to denote the $\alpha$, the $\beta$ and the $\gamma$ that appear in Eqs. (9) through (11) is not very appropriate from my point of view. These are quantities that will depend on the functional form of the PDF $\phi$ in a complex way and, probably, they need a case-by-case study to be assessed.

-I do not understand how Eq. (12) connects with the ACF of the changes of the process. The expression seems dimensionally incorrect. A formal definition of $C(t)$ and the derivation of this equation is needed.

All of the above does not imply that the mathematical developments and the results shown in the manuscript are unreliable or erroneous, but, in my opinion, prevents the manuscript from being accepted in its current form.  

Reviewer 4 Report

The paper addresses interesting model of correlated continuous time random walk (CTRW). The authors suggest specific law of interdependence between waiting times, solve the model analytically and reproduce some statistical properties of high frequency financial time series. The paper serves as a useful addition/continuation of the studies started from the papers [62] and [61]. Of course, the problem of a general theoretical description of CTRW with correlated waiting times is still on the agenda, but the manuscript represents one more step in this direction. For example, how to account for anticorrelations is an open problem to my best knowledge. The paper is clearly written, and I recommend to publish it as is.

Author Response

(The authors gave the same response as above.)

Reviewer 5 Report

To tell you the truth I’m not happy having accepted to review this paper.

Starting from the Introduction, as one should, one is surprised by the vocabulary, style, grammar. Then, one turns toward the (42) mentioned references, in this first Section, and gets astonished. I have absolutely no idea what references 4, 5, 13, 14, 15, 17, 20 are about. Are they part of a search game ? What do I win if I guess the authors ?

Why are refs. 11, 27, 31, written in capital letters ?

Why is there a comment (likely for the authors memory) in some references ?

Sometimes the doi is provided, sometimes not (even for recent papers !)

On the second paragraph, one becomes amazed by the information ; the authors claim that « we can relate quantities of price and volume with each transaction ». That’s unbelievable ; that’s wrong, since the volumes pertain to a sum of transactions and one does not know what price was agreed upon for each specific transaction, and since the trader does not provide the details ; moreover there is some delay between both news.

I read the whole Introduction in order to have some hope that something interesting about Entropy would occur. Nothing. I decided to browse through the paper stopping at a few points which I wished to be of interest. I am flabbergasted.

Only one example : lines 344-5 : « we define extreme event (sic) as an event occurring on average every <N> steps ». This is absolutely not a classical definition of an extreme event !, - not within my domain of expertise!

I give up and recommend that the paper be read again before a new submission to a more appropriate journal.

Round 2

Reviewer 3 Report

I believe the manuscript has been sufficiently improved to warrant publication in Entropy in its present form. With that said, please let me make a final thought on the significance of the model. It is true that, when observed on the tick-by-tick scale, the Wiener process does not provide an accurate description of market behavior. So, if changes are introduced in the laws that govern the evolution of the system, some rationale must be developed, beyond the simple "they fit some plots". If the market does not repeat inter-transaction times, some arguments should be given to support that this may be just an over-simplified description of reality, but a better description at the end: e.g., the deltas associated with the repetitions are surrogates of very peaked distributions sharing the same mean value that are chosen according to psi.